# Differentiable Implicit Solver on Graph Neural Networks for Forward and Inverse Problems

## Abstract

Partial differential equations (PDEs) on unstructured grids can be solved using message passing on a graph neural network (GNN). Implicit time-stepping schemes are often favored, especially for parabolic PDEs, due to their stability properties. In this work, we develop a fully differentiable implicit solver for unstructured grids. We evaluate its performance across four key tasks: a) forward modeling of stiff evolutionary and static problems; b) the inverse problem of estimating equation coefficients; c) the inverse problem of estimating the right-hand side; and d) graph coarsening to accelerate forward modeling. The increased stability and differentiability of our solver enable excellent results in reducing the complexity of forward modeling and efficiently solving related inverse problems. This makes it a promising tool for geoscience and other physics-based applications.

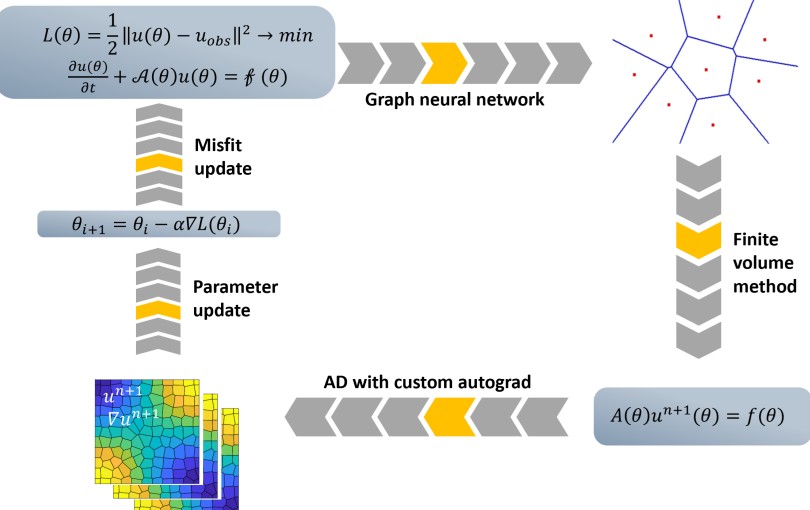

Figure 1: Differentiable modeling pipeline developed in this work combines graph neural networks, the finite-volume method, implicit time-stepping and can be applied to optimization in forward and inverse problems.

## 1 Introduction

In recent years, graph neural networks (GNNs) have emerged as a powerful tool for handling data on unstructured grids (Kipf & Welling, 2017; Veličković et al., 2018; Hamilton et al., 2017; Xu et al., 2019). By leveraging message-passing mechanisms, GNNs are highly effective at capturing local dependencies and updating node features through their connections. This capability has sparked

increasing interest in applying GNNs to PDE solvers, particularly in problems where the data is naturally represented as a graph, such as Voronoi grids and finite-volume discretizations.

Implicit deep learning, El Ghaoui et al. (2021), also referred as deep equilibrium models (DEQ), substitutes conventional activation functions at a particular layers with a non-linear equilibrium equation completed with a *prediction* equation. DEQ combined with GNNs are known as implicit GNNs, Gu et al. (2021). Their benefit is that data propagates instantly within the graph, rather than with finite speed as in case of conventional deep learning.

The finite-volume (FV) method is a well-established numerical technique for solving PDEs, especially on unstructured grids, as it ensures local conservation properties. Integrating FV methods with GNN-based solvers provides a structured, grid-free representation of the domain, while preserving a rigorous framework for solving governing equations. In this context, the development of differentiable solvers is an especially promising research direction.

One key advancement explored in this work is the application of automatic differentiation (AD) Naumann (2012) within GNN-based solvers, particularly for tasks such as graph coarsening and solving inverse problems. AD has proven to be an invaluable tool for efficient gradient-based optimization, especially when combined with deep learning models, enabling end-to-end differentiable pipelines. The use of AD in inverse problems involving hyperbolic equations was demonstrated in Zhu et al. (2021).

In our work, we focus on parabolic partial-differential equations, which are crucial in various applications, including engineering and the oil and gas industry. However, the high stiffness of the resulting evolutionary problems necessitates the use of implicit time-stepping schemes due to their stability properties. Additionally, we address elliptic partial-differential equations, where a distinctive challenge lies in solving large, sparse linear systems.

Although some automatic differentiation (AD) libraries support implicit relations between variables (e.g., JAX), their capabilities for handling sparse linear solvers are currently quite limited. To address this, we developed custom autograd functions tailored to our needs.

Building on this foundation, we propose an integrated approach that combines GNNs, finite-volume methods, implicit time-stepping, and automatic differentiation to create a fully differentiable pipeline for both forward and inverse modeling. Our approach ensures differentiability with respect to various components, including grid cell locations, PDE coefficients, and the right-hand side, providing a flexible and robust framework for tackling both forward modeling and inverse problems. Specifically, we focus on the following key applications:

- Forward modeling of stiff evolutionary and static problems;
- Solving inverse problems to estimate equation coefficients;
- Solving inverse problems to estimate unknown right-hand side terms;
- Graph coarsening to accelerate forward modeling.

We demonstrate the capability of the proposed approach to effectively address all these problem types, showcasing its flexibility and ease of implementation for geoscience and other PDE-based applications.

## 2 METHODOLOGY

### 2.1 GOVERNING EQUATIONS

In this work, we mainly consider a parabolic partial-differential equation in some domain $V$,

$$\frac{\partial u}{\partial t} - \text{div}(K\nabla u) = f, \quad 0 < t < T, \quad (x, y) \in V, \tag{1}$$

where $u(x, y, t)$ is the unknown variable, $K(x, y)$ is some known coefficient, $f(x, y, t)$ is the source term. The above equation is completed with the initial condition $p = p^0(x, y)$ at $t = 0$ and zero Neumann boundary conditions on the domain boundary.

Assume that we given with a set of points $C_s$, $1 \le s \le N$. We use them to generate a Voronoi grid, (1). The a numerical solution to (1) can be received by combining the finite volume method for spatial discretization ((Eymard et al., 2000), (Kuznetsov et al., 2007)) and the implicit Euler scheme for temporal discretization,

$$D\frac{u^{k+1} - u^k}{\tau} + A \cdot u^{k+1} = Df^{k+1}, \quad 0 \le k \le n - 1. \tag{2}$$

where $u^k \in \mathbb{R}^N$, $0 \le k \le n$ is the numerical solution at the respective time step, $A$ is the finite volume matrix, $D$ is the diagonal matrix of Voronoi cells areas, and $f^k$ is the source vector.

We implemented the finite volume method within the graph neural network (GNN) framework with the message passing paradigm. Our GNN solver uses weights for the edges of the graph,

$$w_{ij} = -A_{ij}. \tag{3}$$

Since every non-zero off-diagonal matrix entry corresponds to some edge of the graph, the GNN implements multiplication, $A \cdot u^k$.

## 2.2 PROBLEM SETUP

Optimization in both static and evolutionary forward and inverse problems can be written in the following form,

$$L(\theta) = \sum_{k=1}^{n} \|R \cdot u^k(\theta) - u^k_{obs}\|^2 + S(\theta), \tag{4}$$

where $\theta \in \mathbb{R}^M$ is an unknown parameter requiring estimation, $n$ is the number of temporal measurements (1 for static problems), $u^k(\theta) \in \mathbb{R}^N$ modelled variable, $R : \mathbb{R}^N \to \mathbb{R}^m$ is the measurement operator, $u^k_{obs} \in \mathbb{R}^m$ is measured data at the temporal point $k$, and $S(\theta)$ is a stabilizer. The role of operator $R$ is to simply keep in a given vector several entries (wherever sensors are located) while removing the other entries.

Both static problems and evolutionary problems involving implicit time-stepping will require solution of a large sparse equation system,

$$A(\theta) \cdot u^k(\theta) = f^k(\theta), \quad 1 \le k \le n, \tag{5}$$

forming a constraint to (4).

A naive approach might be to eliminate $u^k(\theta)$ and received an unconstrained optimization problem,

$$L(\theta) = \sum_{k=1}^{n} \|R \cdot A(\theta)^{-1} \cdot f^k(\theta) - u^k_{obs}\|^2 + S(\theta), \tag{6}$$

However, this is not feasible since $A(\theta)^{-1}$ is a large dense matrix not suitable for manipulations including backward propagation. Practically, (5) is solved with either a direct or preconditioned iterative solver.

Let us notice the following. Firstly, automatic differentiation systems that are typically used for minimization of (4), may not process implicit relations like (5), e.g. PyTorch. Secondly, custom developed fast solvers are commonly linked to solve (5). To address both of these cases, we will derive formulae for forward and backward propagations in the next subsection.

## 2.3 FORWARD AND BACKWARD PROPAGATION

We will assume that the system matrix in (5) is symmetric and positive-definite, which typically the case within implicit time-stepping and static problems. Let us rewrite the system as

$$A \cdot u = b \quad \text{or} \quad \sum_{j} a_{ij} u_j = b_i, \tag{7}$$

for brevity. The matrix $A$ is large and sparse thus special formats are used to store its non-zero values only, e.g. compressed sparse row (CSR) or coordinate (COO) formats.

During the forward propagation stage, we have to solve (7), what could be formally written as,

$$u = A^{-1} \cdot b. \tag{8}$$

Practical algorithms avoid storing $A^{-1}$ and multiplication by it.

During the backward propagation stage, we have to compute the gradients of $u$ with respect to $A$ and $b$. The gradient with respect to $b$ could be found as follows,

$$\sum_j a_{ij} \frac{\partial u_j}{\partial b_k} = \delta_{ik}, \quad \text{thus} \quad \nabla_b u = A^{-1}. \tag{9}$$

The gradient with respect to $A$ has the following form,

$$\sum_j \frac{\partial a_{ij}}{\partial a_{km}} u_j + \sum_j a_{ij} \frac{\partial u_j}{\partial a_{km}} = 0, \quad \text{thus} \quad \nabla_A u = -A^{-1} \cdot P \cdot u. \tag{10}$$

where $P$ is a perturbation matrix, i.e. a matrix of zeros with just a single entry of 1.

Now, these expressions can used to compute the gradient of the loss $L$ with respect to $A$ and $b$ efficiently. Notice

$$\nabla_b L = \nabla_u L \cdot \nabla_b u, \quad \text{and} \quad \nabla_A L = \nabla_u L \cdot \nabla_A u, \tag{11}$$

where $\nabla_u L$ is the gradient of the loss with respect to the solution $u$, propagated back from subsequent stages in the computational graph.

Substituting (9) and (10) into (11), we receive,

$$\nabla_b L = \nabla_u L \cdot A^{-1}, \tag{12}$$

and

$$\nabla_A L = -\nabla_u L \cdot A^{-1} \cdot P \cdot u = -\nabla_b L \cdot P \cdot u. \tag{13}$$

The last expressions imply the following computational algorithm the backward propagation stage :

1. Compute $\nabla_b L$ with (12), which is equivalent to solving of a single linear system.

2. Compute $\nabla_A L$ by multiplying the respective entries of $\nabla_b L$ and $-u$ by each other.

Thus computational complexity of the backward propagation stage is equivalent to the complexity solution of a single equation system. Since $A$ is stored in a sparse matrix format, (13) is reduced to differentiating only non-zero entries.

## 3 EXPERIMENTS

### 3.1 IMPLEMENTATION

Our differentiable graph simulator is implemented based on the message passing paradigm in a graph neural networks. For implementing such a solver, we used Pytorch Geometric Fey & Lenssen (2019) framework. Formalas (12) and (13) were implemented in a PyTorch custom autograd function.

### 3.2 FORWARD MODELLING AND ITS ACCELERATION

Let a polygonal domain be filled with heterogeneous porous media. We assume that a slightly compressible fluid is injected at one point and pressure is measured at some other points. With minor simplifications (Chen, 2007) the fluid flow is described by the parabolic equation, (1).

In this experiment, the coarsening process is driven by a graph neural network (GNN), which learns soft cluster assignments to map nodes from the fine grid to coarser clusters. The GNN is composed of graph convolutions followed by an MLP, and it represents a learnable aggregation function $\theta$. The aggregation function $\theta$ learns how to combine the node features from the fine grid into coarse representations, ensuring that key physical properties are preserved during the coarsening process. The feature aggregation is mathematically represented as:

$$\mathbf{c}_j = \frac{\sum_{i=1}^{N} S_{ij}\mathbf{x}_i}{\sum_{i=1}^{N} S_{ij}}, \tag{14}$$

where $\mathbf{c}_j$ is the centroid feature for cluster $j$, $\mathbf{x}_i$ is the feature vector for node $i$, and $S_{ij}$ is the soft assignment of node $i$ to cluster $j$.

The fine modeling grid had 400 cells, Fig. 2 a). We used the time step, $\tau$, of 0.1 s. The pressure was computed at two measurement points. The respective time series are shown in Fig. 2 c). Although the measurement points are symmetric with respect to the source, they are located in the areas of different permeability, leading to different pressure growth rates.

In this experiment, minimization of (4) was performed with Adam optimizer and no stabilizer. We picked a coarse grid with 33 cells and tried to optimize it minimizing the misfit between pressure values on the coarse and fine grids over time. An approximately optimal grid was received after 160 epochs, Fig. 2 b).

The results demonstrate that the coarse grid maintains high fidelity to the original simulation while significantly accelerating the modeling process (from 400 unknowns to 33).

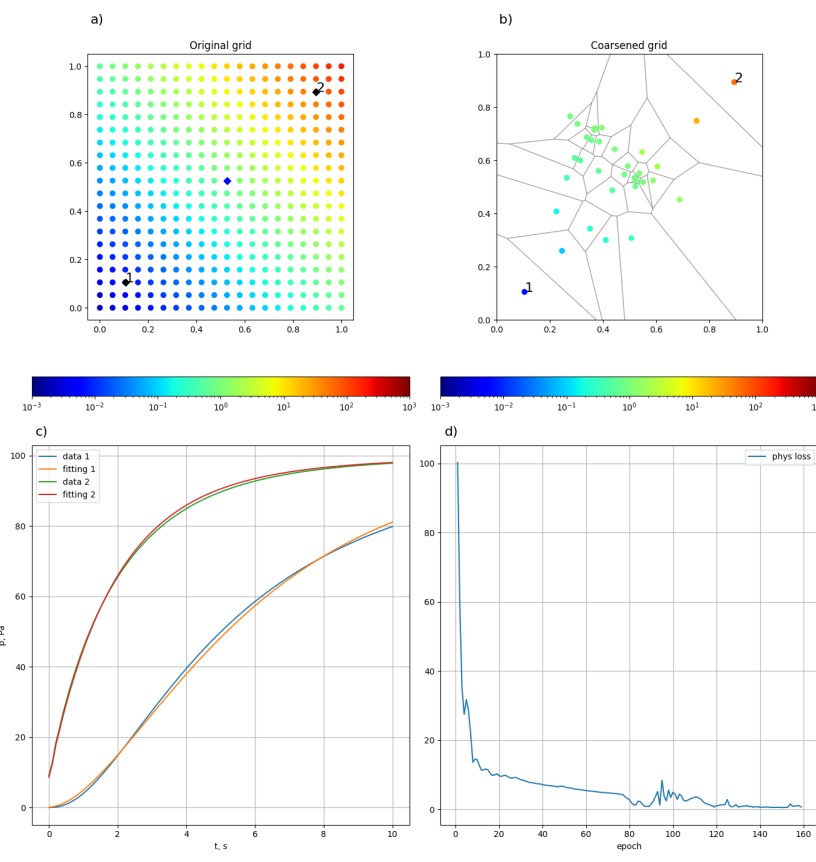

Figure 2: Computational grid coarsening. a) Original grid, color indicates permeability, source marked with blued diamond, and two measurement points (1 & 2) marked as black diamonds; b) coarsened grid; c) original and coarsened grid pressure data at two measurement points; d) loss function during 160 epochs of minimization.

We compared the implicit scheme (2) versus a much easier to implement explicit scheme, Shumilin et al.. The explicit scheme does not involve equation system solution thus computational expenses at every time are quite modest (comparable to matrix-vector multiplication). However, the time step size is limited by a quite restrictive stability condition. Table 1 gives a numerical illustration. We see that although each time step of the implicit scheme is more computationally expensive, the overall

computational time is more than a 1500 time smaller. Fig. 3 shows that the pressure computed with implicit and explicit schemes match quite well.

Table 1: Comparison of implicit and explicit schemes: $\tau$ is the time step size, $n$ is the number of time steps, $T_{cpu}$ is the computational time

| grid | Implicit Scheme | | | Explicit Scheme | | |
|---|---|---|---|---|---|---|
| | $\tau$ | $n$ | $T_{cpu}$ | $\tau$ | $n$ | $T_{cpu}$ |
| 20×20 | 0.01 | 100 | 0.63 | 5e-6 | 2e+5 | 290 |
| 40×40 | 0.01 | 100 | 1.55 | 1e-6 | 1e+6 | 2520 |

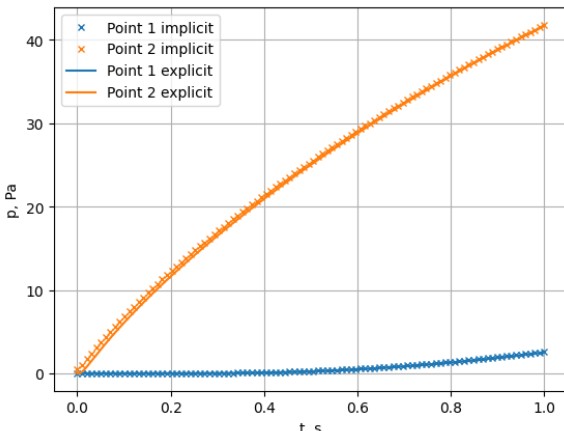

Figure 3: Pressure computed with either implicit or explicit scheme.

### 3.3 COEFFICIENT INVERSE PROBLEM

Estimation of a poorly known partial-differential equation coefficient describing properties of some heterogeneous medium is of paramount importance in many applications, especially geoscience and geophysics.

We consider a setup similar to the previous experiment. However, now we will assume that the true permeability, $K$, is unknown. This problem is known as *history matching* in petroleum engineering. We will estimate the permeability by minimizing the following functional,

$$L(K) = \int_0^T \sum_{i=1}^m w_i(t) \left( p_{mod}^i(t) - p_{obs}^i(t) \right)^2 dt + \beta \int_V |\nabla K|^2 dxdy, \qquad (15)$$

constrained with (1) and $K(x,y) > 0$. Here, the first term represents data misfit, while the second term is a stabilizer promoting permeability smoothness.

In our experiment, pressure was measured at two spatial locations, $m = 2$ and at 100 temporal points, $\tau = 0.1s$, $T = 10s$, Fig. 4 a) and c). The Voronoi grid was formed from a graph of 400 randomly perturbed vertices.

We applied the Adam optimizer to (15) stating with with $K = 1$ and continued for 250 epochs. Decent data fit was archived after 150 epochs, Fig. 4 d). The recovered permeability, Fig. 4 b), has main features of the true one: the lower left area of lower permeability, the upper right area of higher permeability.

### 3.4 INVERSE SOURCE PROBLEM

Another type of an inverse problem that is commonly encountered in geosciences and geophysics is to estimate the right-hand of a partial-differential equation. We will proceed with a setting that

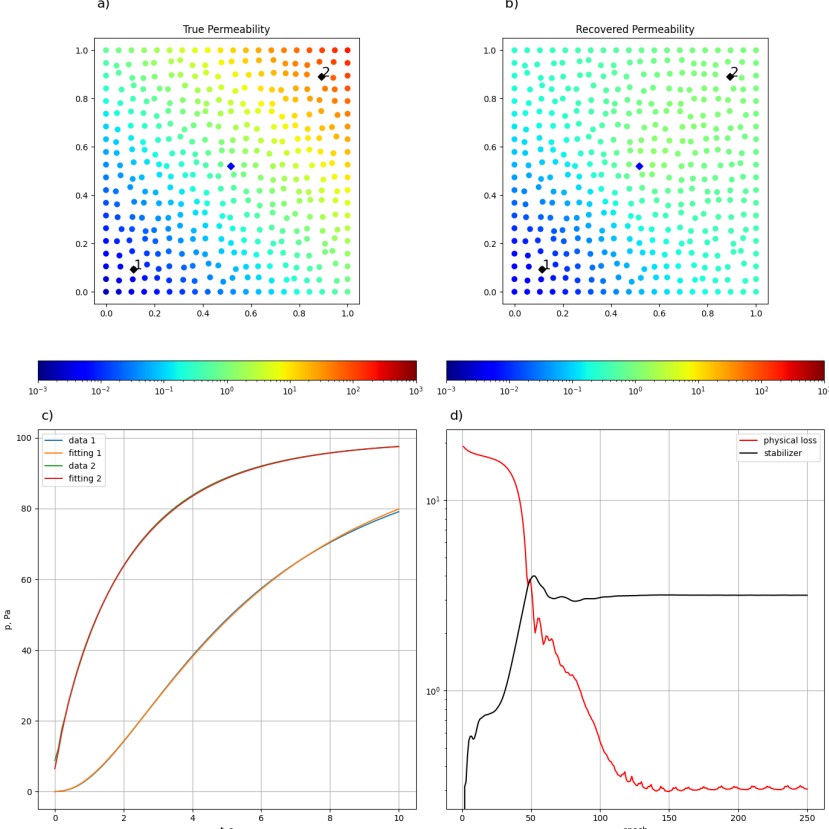

Figure 4: Solution of the coefficient inverse problem. a) Original grid, color indicates permeability, source and two measurement points (1&2); b) recovered permeability; c) pressure at two measurement points (data and prediction/fitting); d) loss function during 250 epochs of minimization.

is referred as *self-potential method*, Revil & Jardani (2013). The goal is to estimate direct current source density based on the observations of the electric potential. They are related by the equation,

$$-\text{div}(\sigma \nabla u) = f. \tag{16}$$

The above equation was considered in a rectangular domain, completed Dirichlet on the bottom side and Neumann boundary on the other three sides. Here $\sigma$ is known electrical conductivity, $u$ is the electric potential, $f$ is source density. Although, the equation is static, its numerical solution will involve solution of a large sparse linear system, making it similar to implicit time stepping. The source density is estimated by minimizing the following functional,

$$L(f) = \sum_{i=1}^{n} w_i \left( u_{mod}^i - u_{obs}^i \right)^2 + \gamma \int_V |\kappa \nabla f|^2 dx dy + \delta \int_V f^2 dx dy, \tag{17}$$

where the first term represents data misfit; the second term is a stabilizer promoting smoothness with $\kappa$ being a variable coefficient needed to compensate decrease of sensitivity with depth; the third term is a stabilizer promoting sources with smaller energy; $\gamma$ and $\delta$ are trade-off scalar parameters.

In this experiment, the Voronoi grid was formed from a graph of 900 randomly perturbed vertices. The electric potential was measured at $n = 28$ points on the upper side of the domain, Fig. 5 a). The true right-hand side was formed by a source in the left part of the domain and a sink in the right part. Observed data, $u_{obs}$, was received by solving (16) and adding 10% noise. Notice this problem was highly under-determined: we estimated the values of 900 unknowns based on 28.

We applied the Adam optimizer to (17) stating with with $f = 0$ and continued for 100 epochs. We observed good data fit and convergence of the optimizer, Fig. 4 c) and d). The recovered and

true right-hand sides, Fig. 4 b) and  a), posses common features: positive values in the left part and negative in the right. The recovered right-hand side is quite smeared. Also the locations of the extrema are biased. These artifacts were related to the measurement setup and impact of stabilizers.  The use of more advanced stabilizers can further improve inversion results.

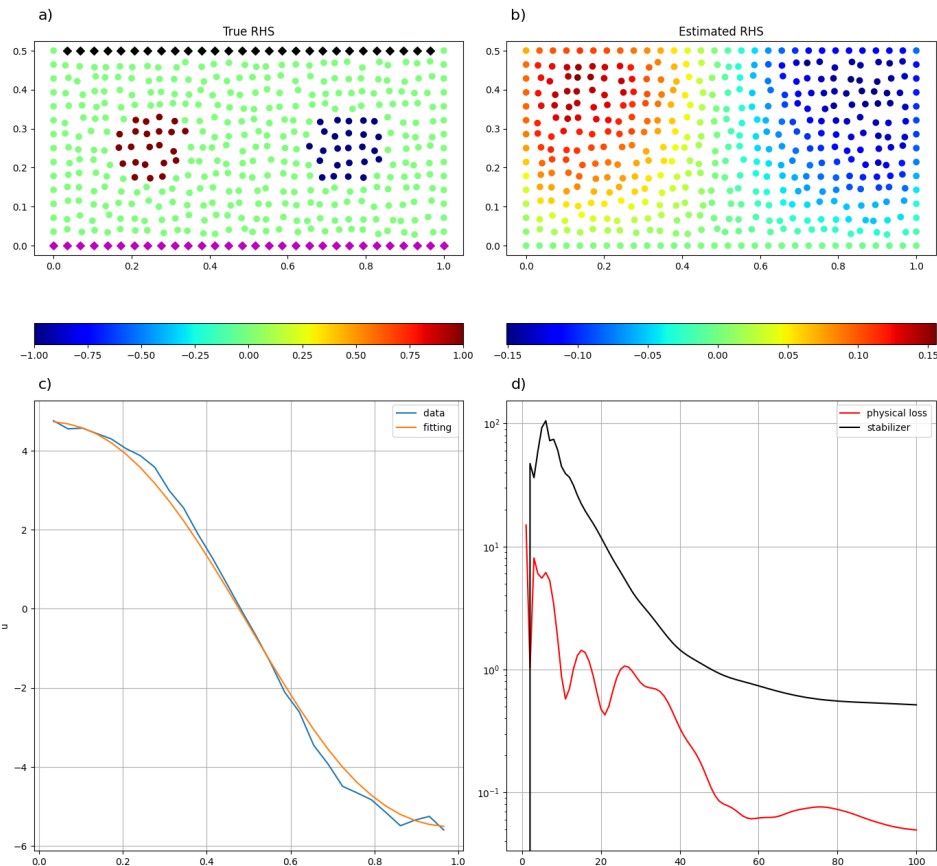

Figure 5: Solution of the source inverse problem. a) True right-hand side and measurement points (black diamonds); magenta diamonds indicate the side where Dirichlet boundary condition was applied; Neumann boundary condition was applied on the other three sides; b) recovered right-hand side; c) observed data at measurement points with with noise added, $u_{obs}$, (data) and prediction, $u_{mod}$, (fitting); d) loss function during 100 epochs of minimization: physical loss (first term in (17)) and stabilizers (later terms in (17)).

## 4 RELATED WORK

A large number works of recent works can be characterized as surrogate modelling. For forward problems, the idea idea it approximate the equation Green's function, e.g., Alet et al. (2019) or Nastorg et al. (2024) involving implicit GNN. For inverse problems, the idea is approximate the inverse to the possibly non-linear forward operator using a given dataset. For example, Zhao et al. (2022) involved GNN for this task. In Jessica et al. (2023), the author enriched GNN with novel geometric features for better prediction accuracy. The work of Horie & Mitsume (2024) suggested a learnable approach to compute fluxes in the FVM, motivated by convection-dominated problems. However, these approach typically requires massive datasets (GBs and TBs) and quite computationally demanding learning (100-1000s of GPU hours).

In the traditional approach to inverse problems, Zhdanov (2002), optimization on performed every new data. But this algorithms are implemented though derivation of the Jacobians. This derivations a tedious and prone-to-bugs task. Availability of AD pipeines can help to avoid it. The work of Zhu

et al. (2021) presented AD-based acoustic data inversion combined with explicit Euler temporal discretization of the wave equation. In our work, we followed a similar direction and showed how the implicit schemes can be incorporated into an AD pipeline and further combined with a GNN. Our work is close to PINNs, Raissi et al. (2019), Lu et al. (2021), and their extensions, graph convolutional PINN, Gao et al. (2022), since both of the approaches can be attributed as self-supervised.

## 5 CONCLUSIONS

In this work, we integrated graph neural networks, the finite-volume method, and implicit time-stepping to develop a fully differentiable modeling pipeline, applying it to both forward and inverse problems in geoscience. Our PyTorch-based pipeline is flexible and easy to implement, enabling the integration of new, fast solvers for large sparse linear systems.

For comparison, Shumilin et al. presented results on accelerating forward modeling using graph coarsening, which is somewhat similar to our approach in Section 3.2. However, their method encountered significant time-step limitations due to the use of an explicit scheme.

Our pipeline is highly efficient. In future work, we plan to address large-scale 3D problems and more complex equations, such as multi-phase fluid flow. Large-scale 3D modeling and inversion typically rely on custom preconditioned iterative solvers, as they offer both memory efficiency and fast convergence. Our pipeline's ability to easily integrate new solvers makes it particularly appealing for industrial applications.

In this work, we discussed coarsening within the context of forward modeling and data inversion. An interesting future research direction would be to combine coarsening with inversion, allowing data inversion to be performed on an optimal coarse grid.

## 6 REPRODUCIBILITY

Our implementation mainly follows the formulas derived in the manuscript. The source code is available upon written request.

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

# A   CONSERVATION PROPERTY

In this appendix, we derive an expressions for energy balance and illustrate energy conservation numerically.

Let us integrate (1) over domain $V$. With help of the divergence theorem, we arrive to the following,

$$\frac{\partial}{\partial t} \int_V u dV = \int_V f dV. \tag{18}$$

This equality states that source energy fully is converted to the change of solution energy.

The FVM allows to receive an identical discrete equality. Multiplying (2) by vector $e = (1 \cdots 1) \in \mathbb{R}^N$, we arrive to,

$$\frac{1}{\tau}(e, D(u^{k+1} - u^k)) + (e, Au^{k+1}) = (e, Df^{k+1}). \tag{19}$$

However, $A$ is singular, thus the later expression is simplified to

$$\frac{1}{\tau}(e, D(u^{k+1} - u^k)) = (e, Df^{k+1}). \tag{20}$$

We see that FVM variables obey the same conservation property as the contentious ones.

Given the setup shown in Fig 4a), we made a plot to illustrate (20). Fig. 6 shows total source energy and change of the solution energy at every time step. We observe that they are match identically.

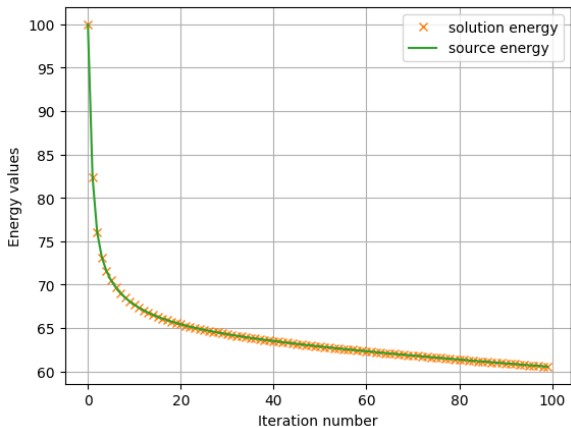

Figure 6: Total source energy (green line) and change of the solution energy (crosses).

