# OpenReview forum: "Differentiable Implicit Solver on Graph Neural Networks for Forward and Inverse Problems"
_ICLR.cc/2025/Conference — Submitted to ICLR 2025_

### Official Review · Reviewer_2K7B · 2024-11-02

**Soundness:** 2
**Presentation:** 1
**Contribution:** 1
**Rating:** 1
**Confidence:** 3

**Summary:**

This paper explores the use of graph neural networks for solving forward and inverse problems and particularly focuses on the incorporation of implicit solver. However, the writing is subpar and the procedures and advantages are not well explained. The experiments are also lack comparison with other methods.

**Strengths:**

The question is interesting and combining GNN with finite element method seems natural.

**Weaknesses:**

1. The writing is subpar. There are many typos and grammatical errors. For example, "Compute $\nabla_bL$  with (12), whats is equivalent so the solution of a single linear system." should be "Compute $\nabla_bL$ with (12), which is equivalent to solving a single linear system."
2. One main focus of this paper is the incorporation of implicit solver. However, using an iterative solver and in a deep learning setting is well-studied in the Deep Equilibrium Models (DEQ) literature. The authors should compare their method with DEQ.
3. The experiments are not very convincing. The results in Section 3.4 is very poor and in no experiments the authors compare their method with other methods.

**Questions:**

See weakness

---

> ### Author Response · Authors · 2024-11-29
>
> **The writing is subpar**
>
> We appreciate the reviewer for this remark. This and other typos/errors were corrected.
>
> **The authors should compare their method with DEQ.**
>
> In our work, we discuss implicit constraints for optimization problems.
> While DEQ is based on special activation maps (rather than recursive functions in conventional deep learning).  We don’t see a relation between these two notions.
>
> **The results in Section 3.4 is very poor and in no experiments the authors compare their method with other methods.**
>
> We redid our experiments in Section 3.4 with a new cost function, which ultimately improved inversion results.  Please review this section in the updated manuscript.
> See also Official Comment #2 above.

---

### Official Review · Reviewer_QrEY · 2024-11-04

**Soundness:** 1
**Presentation:** 1
**Contribution:** 2
**Rating:** 3
**Confidence:** 4

**Summary:**

This paper proposes an integrated approach for solving forward and inverse problems by creating a new pipeline that combines Graph Neural Networks (GNNs) with Finite Volume Methods (FVM) to enable automatic differentiation with implicit-time stepping.

**Strengths:**

Figure 1 effectively illustrates the overall pipeline, demonstrating experimental results that apply the combination of GNN and FVM to both forward and inverse problems.

**Weaknesses:**

First and foremost, the paper feels incomplete. The biggest concern is the lack of discussion about other approaches that use GNNs or integrate FVM with deep learning to solve PDEs. A “Related Work” section should be added to explain how the proposed model differs from recent studies and highlight its novelty. Although Section 2 on theory explains the problem setup to some extent, more detailed steps and methods for training the proposed approach should be included. Section 3, the experimental part, merely lists the results for forward and inverse problems without discussing how this method compares to existing GNN- and FVM-based approaches. For instance, the study "Learning to Solve PDE-constrained Inverse Problems with Graph Networks" solves inverse problems using GNNs—how does the proposed method differ from this approach, and what advantages does it offer? Experimentally, does it outperform in solving inverse problems?

**Questions:**

* Why should we only consider the Neumann boundary in equation (1)? Is it difficult to consider other Robin boundaries?

* I don't understand the role of R in equation (4). What does it mean as a measurement operator?

* Does S(theta) change depending on the equation of the PDE to be solved? Can you explain this further?

* In equation (8), we need to find A^{-1} in the end. Isn't the cost for this large?

---

> ### Author Response · Authors · 2024-11-29
>
> **First and foremost, the paper feels incomplete**
>
> We appreciate the reviewer for these remarks.
> Our approach is self-supervised, thus massive datasets are not needed;
> Training could be performed with any optimization method, either stochastic or deterministic.
> The work  "Learning to Solve PDE-constrained Inverse Problems with Graph Networks" is not feasible for comparison since it involves solving the wave equation. For solving the wave equation explicit schemes are totally sufficient.  Incorporation of explicit schemes into the AD framework was earlier discussed by Zhu el, 2021.
> In contrast, our work is focused on incorporation of implicit schemes. We thus did not include the wave equation in our experiments.
>
> We added a Related Works Section and discussed the originality of our approach there.
>
> Also See Official Comment #2 above for a comparison versus published works.
>
> **Why should we only consider the Neumann boundary in equation (1)? Is it difficult to consider other Robin boundaries?**
>
> There is no limitation on boundary condition types since we follow the traditional FVM.
> For example, experiments in Section 3.4 Inverse Source Problem involve both Dirichlet and Neumann boundary conditions.
>
> **I don't understand the role of R in equation (4). What does it mean as a measurement operator?**
>
> FVM provides the solution within the whole modelling domain. However, in many applications we have data access only at one, two or several points of the domain (e.g. the humans typically measure temperature at one point; earth subsurface properties are known only where a well was drilled).  Operator R simply keeps in a given vector several entries (wherever sensors are located) while removing the others. In Figs 2a), 4a), 5a) these points are indicated as black diamonds.
>
> **Does S(theta) change depending on the equation of the PDE to be solved? Can you explain this further?**
>
> A significant portion of inverse problems are ill-posed thus admitting an infinite number of solutions. A stabilizer S(theta) is intended to regularize an ill-posed problem, making it well-posed. The choice of S(theta) depends not only on equation type, but also on physical properties of the unknown variable as well as the measurement setup.
> For example, the reader may notice that different stabilizers were used in (15) and (17).
>
>
> **In equation (8), we need to find A^{-1} in the end. Isn't the cost for this large?**
> Eq (8) gives only an expression for $u$. In our implementation, we solve $A u=b$ to find $u$ with a sparse direct method, thus manipulations with $A^{-1}$ are avoided.
>
> Zhu et al,  A general approach to seismic inver-sion with automatic differentiation. Computers and Geosciences, 151:104751, June 2021. ISSN 0098-3004. doi: 10.1016/j.cageo.2021.104751. URL http://dx.doi.org/10.1016/j.cageo.2021.104751

---

### Official Review · Reviewer_1yfg · 2024-11-04

**Soundness:** 2
**Presentation:** 1
**Contribution:** 1
**Rating:** 1
**Confidence:** 3

**Summary:**

The work considers mesh coarsening, forward, and inverse problems and investigates the implicit solver. In the numerical experiments, the authors evaluated the performance of the method regarding each problem.

**Strengths:**

* The work tries to build a framework that works with mesh coarsening, forward, and inverse problems.

**Weaknesses:**

* The novelty of the work is limited. Incorporating FVM into GNN is not new and considered in, e.g., [Jessica et al. ICML 2024 https://arxiv.org/abs/2311.14464 ] and [Horie et al. ICML 2024 https://arxiv.org/abs/2405.16183v1 ]. The construction of gradients presented in Section 2.3 seems strongly related to the adjoint method, which is a standard way to deal with inverse problems. The implicit method for GNN is considered in the area of implicit GNNs, e.g., [Gu et al. NeurIPS 2020 https://arxiv.org/abs/2009.06211 ]. The authors state that these are their novelty, but there is existing work for each. The authors should cite these works and clarify the added novelty from the authors.
* The evaluation is weak. There is only one baseline for the experiment in Section 3.2 and nothing for the ones in Section 3.3 and 3.4. With the current form, the reviewer cannot asses the effectiveness and superiority of the model.
* The presentation is not clear. The figure may miss the labels (a), (b), and so on for Figures 2, 3, and 4. It is not clear what is "data 1", "fitting 1", "data 2", and "fitting 2" in Figures 2 and 3.

**Questions:**

* What would be the limitation of the method?
* What would be the potential benefit of using machine learning for linear PDE over classical methods?

---

> ### Author Response · Authors · 2024-11-28
>
> **The novelty of the work is limited**
>
> We appreciate this remark.
> We added a Related Works Section and discussed the originality of our approach there.
> Backpropagation and the adjoint state method are known to be equivalent,
> see e.g. Backpropagation: Theory, Architectures, and Applications, Psychology Press, 2013.
>
>
> **The evaluation is weak. There is only one baseline for the experiment in Section 3.2 and nothing for the ones in Section 3.3 and 3.4.**
>
> We added a comparison, see Table 1. See Official Comment #2 above for a comparison versus published works.
>
> **The presentation is not clear...**
>
> We have updated the plots for better readabilty.
>
> **What would be the limitation of the method?**
>
> Resolution of some inverse problems is quite low (e.g. due to a measurement setup). It is thus not feasible to represent the unknown variable on a near-uniform grid (Figs 4b and 5b). In our future work, we will investigate leanable grid coarsening within inverse problems.
>
> Currently we are using a SciPy direct solver for solution Eqs (7) and (12). However, large-scale modelling and inversion would require a more economical iterative solver, which will be implemented in the future.
>
> **What would be the potential benefit of using machine learning for linear PDE over classical methods?**
>
> This is a quite general question actually.
> The particular problem that we addressed in our work is the incorporation of implicit constraints into the AD framework. Deriving jacobians for classical numerical methods is a colossal and prone-to-bugs task. Here it is avoided.
> Specifically, We designed a fully differentiable implicit solver for a linear PDE that can be used within larger pipelines, e.g. solving inverse problems with gradient methods.

---

### Official Review · Reviewer_4Kqz · 2024-11-04

**Soundness:** 1
**Presentation:** 1
**Contribution:** 2
**Rating:** 3
**Confidence:** 3

**Summary:**

This paper introduces a novel framework that combines graph neural networks with the finite volume method, to address implicit schemes. (There was a challenge that differential equation solvers typically avoid due to the additional computation complexity of handling implicit equations.)

**Strengths:**

By employing an implicit scheme with optimized gradient computation, the proposed method reduces the required number of time steps.
They present a differentiable framework for both forward and inverse methods, enabling a learnable numerical approach based on discrete time steps.
Additionally, the paper explores applications in inverse problems, often employing irregular unstructured grids as used in practical scenarios.

**Weaknesses:**

While the underlying idea is promising, the paper would benefit from stronger experimental or theoretical justification for the proposed methodology. Additional clarity and motivation for the approach would enhance the paper’s impact.

**Questions:**

1. In line 74, the authors discuss the limitations of automatic differentiation in JAX. Further elaboration on this limitation would improve the motivation for this approach.

2. Graph neural networks are frequently used to manage unstructured grid points. Since the paper emphasizes integration with the finite volume method with its local conservation property in line 58, it would be beneficial to include experiments validating these conservation properties.

3. Equation (12) appears to involve matrix inversion on the right-hand side of the proposed gradient formulation. Could the authors address whether this matrix inversion contributes to computational costs, comparable to previous methods?

4. In Equations (12) and (13), gradient computations are proposed. An alternative approach might involve solving the implicit equation through optimization techniques commonly used in deep learning, such as constructing a minimization problem for Equation (5) combined with a data loss function, potentially avoiding matrix inversion. Could the authors discuss this approach?

5. While the finite volume method can accommodate various boundary conditions, the paper considers only Neumann boundary conditions. Is there a specific reason for this choice?

6. In line 253, the authors claim lower computational costs for their method than the explicit Euler scheme, which requires smaller time steps. Could the authors provide a detailed comparison of the computational cost per each time step to support this claim?

7. In Figure 2, the initial scale of the loss is relatively high, making it difficult to assess whether the loss converges to zero after 160 epochs. Given that a coarser grid, intended to reduce computation, may negatively impact estimation accuracy even if the loss function converges to zero, a guideline for determining sufficient loss minimization would be beneficial.

8. In Figures 3 and 4, the recovered permeability only captures general trends rather than precise values. However, the proposed method in (c) accurately approximates the true data distribution, which suggests that the problem may be inherently ill-posed, where the coefficient may not be unique in this setting. Could the authors clarify whether this issue arises from the problem or the numerical method?

9. All experiments utilize a large number of data points, which may facilitate finding a solution. Additional experimental results with fewer grid points would strengthen the paper.

---

> ### Author Response · Authors · 2024-11-28
>
> **the paper would benefit from stronger experimental or theoretical justification**
>
> We extended the Into and Experimental sections and added Related Works Section
>
> **limitations of automatic differentiation in JAX**.
>
> The JAX AD solver is based on the QR matrix factorization. It is suitable for small and moderate size problems, but for larger problems it will be too slow and memory-consuming.
> In our work, we show how an arbitrary solver could be integrated into the automatically  differentiationable framework.
>
> **validation of conservation properties**
>
> We added the respective experiment to the manuscript, Appendix A.
>
> **Equation (12) appears to involve matrix inversion**
>
> Eq. (12) should be interpreted as solution of a linear,
> $A \cdot \nabla_b L = \nabla_u L,$
> with a  known right-hand side,  $\nabla_u L$.
> The computational cost of this step is equal to the complexity of solving a sparse linear system.
>
> **solving the implicit equation through optimization techniques**
>
> Matrices arising from discretization of elliptic PDEs, like (5), are highly ill-conditioned. Thus incorporation of such a matrix intro optimization problems would lead to very slow convergence of the minimization algorithms (both stochastic and deterministic).
>
> **various boundary conditions**
>
> Experiments in Section 3.4 Inverse Source Problem involve both Dirichlet and Neumann boundary conditions.
>
> **provide a detailed comparison with explict Euler**
>
> We added an experiment on this topic. See Table 1 in the revised manuscript.
>
> **In Figure 2, the initial scale of the loss is relatively high, making it difficult to assess whether the loss converges to zero after 160 epochs.**
>
> The initial loss might be scaled weights w_i, Eq (15), so that it is 1.
> So far we followed the standard ML  guideline  to stop optimization wherever the loss is sufficiently flat. But a more precise criteria is in fact needed.
>
> **coefficient may not be unique in this setting. Could the authors clarify whether this issue arises from the problem or the numerical method?**
>
> Both problems in Fig 4 and 5 are highly under-determined. For example in Fig 4, we estimated the values of 900 unknowns based on 28. This implies non-uniqueness of the solution of the inverse problem.
>
> **Additional experimental results with fewer grid points would strengthen the paper.**
>
> We have not managed to perform such an experiment with the given time frames. But we will evidently address this topic in the future.

---

### Author Response · Authors · 2024-11-28

Dear Reviewers,

We appreciate your remarks.
We should emphasize that our work focuses on incorporation of implicit constraints into the AD framework.
The main difference versus several main-stream papers is that
* our approach is self-supervised, thus massive datasets are not needed;
* our approach does not produce a surrogate solver, but rather follows the FVM discretization respecting energy conservation.
* In particular, our approach is quite close to the traditional method of solution of inverse problems, i.e. Tikhonov functional minimization. However, we employed AD to avoid derivation of jacobians and GNN for unstructured data representation. The challenge was to implement AD efficiently and flexibly whenever implicit constraints are given and described by large sparse matrices

We revised the manuscript to address the issues and added new experiments.
The new text is in red for your convenience.

If there are any further questions or points that need a discussion, we will be happy to address them.

 Best regards,

 The Authors

---

> ### Author Response · Authors · 2024-12-03
> **Official Comment #2**
>
> Dear Reviewers,
>
> As suggested, we performed a comparison with a baseline method.
>
> Continuing Section 3.4, Inverse Source Problem, we compared the results of our inversion method with those obtained using physically-informed neural networks (PINNs) [1], implemented via the Deepxde package [2]. In PINNs, both the unknown solution and the right-hand side are parameterized by a neural network. For our example, we employed a fully-connected neural network with two inputs, three dense layers (150 nodes each), two outputs, and a sine activation function. The same setup described in Section 3.4 was analyzed for consistency. Both minimizations were performed using the Adam optimizer.
>
> The figure below illustrates the results. The PINN successfully captured the primary features of the right-hand side (e); however, the source and sink were inaccurately located near the upper boundary of the domain (measurement line). Our approach was somewhat more accurate in recovery of the source and sink (b): there is a displacement above their true locations. Moreover, our method more precisely captured the magnitudes of the source and sink, than the PINN.
>
> Lastly, we note that the convergence of the optimizer in our method was significantly faster, as shown in (d) and (f). We attribute this difference to the highly nonlinear nature of the loss functions in PINNs, whereas our approach involves a quadratic loss function (eq. 17), leading to more efficient optimization.
>
>
> [**Figure**](https://drive.google.com/file/d/1nUIZx8oDFwewCOknFYTRdyPMMDwI9465/view?usp=sharing)
>
>
> [1] Raissi et al, Physics-informed neural networks: A deep learning framework for solving forward and inverse problems involving nonlinear partial differential equations. Journal of Computational Physics, 2019.
>
> [2] Lu et al,   A deep learning library for solving differential equations , SIAM Review, 2021.

---

> ### Author Response · Authors · 2024-12-03
> **Official Comment #3**
>
> Dear Reviewers,
>
> We appreciate your constructive feedback. We would like to emphesize changes that we made in the revised manuscript.
>
> * Added Related Work Secttion
> * We added an experiment on comparison of implicit and explicit time-stepping. See Table 1 in the revised manuscript
> * We added an experiment illustrating energy conservation, see Appendix A.
> * We have updated the plots for better readability.
>
> We also made a comparison versus the PINN (see Official Comment #2), which we plan to include in the camera-ready version.
> We hope all the reviewers will consider raising their scores further, taking into account the extensive improvements made to the paper and the additional experiments included.
>
> Best regards,
>
> The Authors

---

### Meta-Review · Area_Chair_VPPD · 2024-12-20

**Metareview:**

(a) The high level idea pursued in this paper is to combine finite volume methods, graph neural networks (GNNs), and implicit time-stepping to solve partial differential equations on unstructured grids. (b) The idea is intuitive but (c) is supported by a limited set of experiments on toy datasets. (d) Scientific machine learning is now a rather mature sub-field, and in my opinion any new techniques, even if intuitive,  should be supported with solid evaluations on a diverse set of PDE systems, forcing functions, boundary conditions, etc. This paper falls (significantly) short in this regard. The original version of this paper also didn't have any discussion about connections with related work.

**Additional Comments On Reviewer Discussion:**

The authors responded by adding a few more experimental results along with a (short) discussion of related work. This is a good start but much more needs to be done in this direction, particularly on the evals front.

---

### Decision · Program_Chairs · 2025-01-22

Reject